# Reaching articular limits can negatively impact embodiment in virtual reality

Thibault Porssut[1,2,3]*, Olaf Blanke[3], Bruno Herbelin[3], Ronan Boulic[2]

**1** Altran Lab, Capgemini Engineering, Paris, France, **2** Immersive Interaction Research Group, École Polytechnique Fédérale de Lausanne (EPFL), Lausanne, Switzerland, **3** Laboratory of Cognitive Neuroscience, Brain Mind Institute, École Polytechnique Fédérale de Lausanne (EPFL), Lausanne, Switzerland

* thibault.porssut@epfl.ch

**Data Availability Statement:** All the files are available from Zenodo. The DOI is 10.5281/zenodo.5986673.

**Funding:** T.P This work has been supported by the SNFS project 'Immersive Embodied Interactions'

## Abstract

Providing Virtual Reality(VR) users with a 3D representation of their body complements the experience of immersion and presence in the virtual world with the experience of being physically located and more personally involved. A full-body avatar representation is known to induce a Sense of Embodiment (SoE) for this virtual body, which is associated with improvements in task performance, motivation and motor learning. Recent experimental research on embodiment provides useful guidelines, indicating the extent of discrepancy tolerated by users and, conversely, the limits and disruptive events that lead to a break in embodiment (BiE). Based on previous works on the limit of agency under movement distortion, this paper describes, studies and analyses the impact of a very common yet overlooked embodiment limitation linked to articular limits when performing a reaching movement. We demonstrate that perceiving the articular limit when fully extending the arm provides users with an additional internal proprioceptive feedback which, if not matched in the avatar's movement, leads to the disruptive realization of an incorrect posture mapping. This study complements previous works on self-contact and visuo-haptic conflicts and emphasizes the risk of disrupting the SoE when distorting users' movements or using a poorly-calibrated avatar.

## Introduction

Full-body avatar representation in VR has been shown to be beneficial for user engagement and for the efficacy of a VR simulation [1]. The subjective experience users have when feeling embodied inside their avatar is called the Sense of Embodiment(SoE). According to Kilteni et al. [2], SoE is composed of three main components; body ownership, agency, and self–location. As defined by Lenggenhager et al the self-location is the determinate volume in space where one feels to be located. Normally self-location and body-space coincide in the sense that one feels self-located inside a physical body [3]. The sense of agency refers to the sense of having "global motor control, including the subjective experience of action, control, intention,

grant 200020_178790. Swiss National Science Foundation: https://www.snf.ch/en NO: The funders had no role in study design, data collection and analysis, decision to publish, or preparation of the manuscript.

**Competing interests:** NO: The authors have declared that no competing interests exist.

motor selection and the conscious experience of will" [4]. Body ownership refers to one's self-attribution of a body [5, 6].

Numerous studies [7–10] demonstrated that the subjective experience of embodying an avatar is not limited to a specific technological implementation, but rather emerges from the appropriate conjunction of cognitive processes. As such, a simplified but carefully designed avatar animation can be sufficient to lead to a sufficient level of SoE (e.g. motion capture of the two hands only, with inverse kinematics (IK) for the rest of the body), but only to the point when the errors linked to this implementation are affecting the subjective expectations of the participants (e.g. non-tracked feet not following the participants' movement can be highly disturbing during locomotion). Disrupting embodiment, with the occurrence of a Break in Embodiment(BiE) [11], can indeed be worse than not having a virtual body at all [12]. Furthermore, specific tracking errors causing visuo-proprioceptions conflicts have a definite impact on people's sense of embodiment and performance [13].

In this context, we observed a relatively frequent problem likely to interrupt the visuo-proprioceptive integration, and thus be noticed by participants: when participants are fully extending the arm, they perceive the articular limit as an additional proprioceptive information. Perceiving this internal haptic feedback raises awareness on participant's arm postures which, if it is not matched with a fully extended virtual arm posture, can potentially break embodiment. Such a posture mismatch could, for instance, originate from the morphological differences between participants or from an imperfect avatar calibration. This visuo-proprioceptive conflict might also occur when the avatar animation algorithm is causing a posture difference or due to tracking errors. Ferrell and Smith [14] studied the role of joint receptors in signaling joint movement. They demonstrate that joint receptors work only in the extreme motion range, acting as "limit detectors" helping to define the limits of limb movement (see review [15]), or in other words that the articular limit acts as an internal haptic feedback. Given that subjects may rely more on proprioception to compensate the reduced precision of the visual feedback in depth, perceiving this specific articular limit may weigh significantly in the precision of the movement. Moreover, if users get any extra feedback when reaching articular limit, it may even dominate over vision. This is comparable to the work of Bovet et al. [16] who showed that an additional self-haptic feedback results in visuo-tactile and visuo-proprioceptive mismatches, that strongly disrupt subjects' embodiment.

The impact of this internal haptic feedback is however not limited to an algorithmic problem of motion capture. The benefit of full body embodiment indeed comes from the flexibility for mapping real movements of the user into virtual movements of their avatar, and introducing artificial distortions helps in resolving other frequent VR conflicts, such as when displacing the avatar's hand to avoid going through virtual objects. The most common approach for movement distortion is to introduce a distortion to the location of the virtual hand and assessing to what extent subjects are tolerant of the introduced discrepancies [17–19]. These studies could identify a threshold, called a *detection threshold*, under which participants do not perceive that a distortion is applied between the apparent movement of the avatar (seen in the first-person perspective in VR) and the actual movement they performed. Interestingly, Ogawa et al. [20] find a significant effect of the visual appearance on the distortion threshold in the leftward direction. They hypothesize that it might be due to the difference in muscle execution. When reaching the left position, subjects were in flexion, contrary to the right position where they were fully extended. Their results might however be explained by this additional internal haptic feedback and not by the sole visual appearance. However, when a strong sense of agency coupled with a synchronous visuo-tactile correlations is provided, the mismatch conflicts may be decreased [21].

This study evaluates whether reaching the articular limit when fully extending the arm would make the subjects aware of a movement distortion which was otherwise unnoticed. We anticipate that the internal haptic feedback is strong enough to make users aware of a discrepancy between the virtual and the real arm postures, making it easier for subjects to notice a distortion. To investigate this, we systematically and purposefully provoke this disruption during a reaching movement and evaluate its impact on the SoE.

We first hypothesize that the distortion has a bigger negative effect on the detection threshold when the distortion hinders the subjects' movement (negative discrepancy) than when it helps (positive discrepancy)(H1). Our second hypothesis is that the detection threshold is lower when an articular limit is reached than when not (H2). Extending these hypotheses to the impact of our manipulation on the subjective experience of embodiment, we also hypothesize that, with the same distortion value, subjects will more often reject the virtual body for a negative discrepancy than for a positive discrepancy (H3). Similarly, our last hypothesis is that the detection of an articular limit has a negative impact on the sense of embodiment (H4).

## Materials and methods

### Participants

The experiment was conducted with twenty-five subjects (18 to 31 years old, 21.14 ± 1.9, nine females, one left-handed). Even though the task was not difficult, not controlling the dominance laterality could be considered as a limitation of the study. Indeed, the joint receptors of the participants might be more sensitive with a higher proprioceptive sense in the dominant upper limb compared to the non-dominant upper limb.

Before starting the experiment, subjects were asked to read the information sheet and complete the consent form. Then they needed to fill in a form with questions about their background (gaming experience, previous experience with VR applications). Participants were compensated for their participation. One subject's data was discarded due to technical issues. The study was approved by our local ethics committee and performed in accordance with the Declaration of Helsinki.

### Equipment and software

Participants were equipped with a Pimax 5K Plus, a Head-Mounted Display (HMD) with 2,560 x 1,440 pixels per eye at 90 Hz refresh rate and weighing 514 g. It has a field of view (FOV) of 200° (diagonal) with 170° horizontal and 115° vertical FOV. This choice ensured that the elbow was always inside the FOV when performing the task, thus ruling out the visibility of the arm as a potentially confounding factor. Bose QuietComfort 35 wireless headphones with active noise canceling were used to play a non-localized white noise during the experiment, and only interrupted when communicating with the subject. Seven HTC Vive Trackers V2 were used to track the different body parts (chest, shoulders, elbows, and hands). An HTC Vive Controller, held in the left hand, was used to answer questions (button press).

The virtual environment was a square room of $6 \times 6 \times 3 m^3$ with a chair in the middle. An avatar holding a black cylinder in the right hand was calibrated to collocate with the subject's body. Subjects held an actual physical cylinder. This cylinder was not tracked, its position and orientation were computed from the hand orientation and position. Holding object ensured a visuo, proprioceptive, and tactile coherence between the real and virtual hands in the absence of finger tracking. The application was implemented using Unity 3D 2019.2.0f1.

Subjects were seated for the whole duration of the experiment and performed simple movements with their right hand. They saw their virtual avatar in first person view. The posture of the avatar was reconstructed with the analytic Inverse Kinematic solver (IK) to correspond to

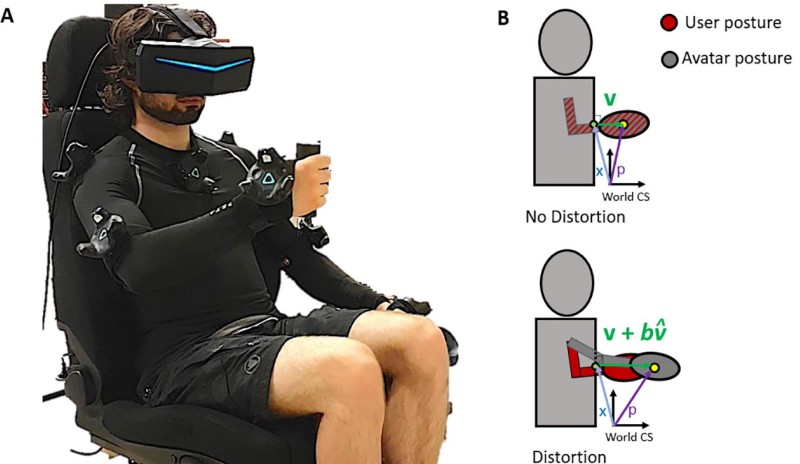

**Fig 1.** Left: Equipment: Participants have trackers on shoulders, elbows, and both hands, as well as a tracker on their chest. They hold a cylinder in the right hand and an HTC Vive Controller in the left hand. Right: Distortion of the subject's hand position. No distortion: avatar and subject's real posture coincide; Distortion: avatar's posture. $v$ is the relative displacement vector to the body part surface; $\hat{v}$ the normalized displacement vector; $b$ is the offset position between the real and virtual hand (5.5 cm in our experiment).

the posture of the subject using LimbIk from FinalIK (root-motion.com) package. The avatar was first scaled based on the subject's height. Then the shoulder width and the trunk length were automatically calibrated based on the shoulder tracker position. The length of the arms and the position of the hands were calibrated during this step as well. The avatar was not visible during the calibration to prevent the subject from viewing visual artefacts.

## Distortion model

This study uses a body-centered distortion model that co-locates the real and the virtual hands whenever the real hand is in contact with the torso, and that allows introducing an offset (amplifying or hindering) to the torso-hand vector in order to implement a movement distortion (see Fig 1). More details on the implementation of the distortion algorithm can be found in Bovet et al. [16].

**Body coordinate system.** Hand movements are expressed in a reference frame relative to the torso, which is simplified to a parallelepipedic shape. The position of the subjects' virtual hand is computed relatively to the torso as shown in Eq 1. The position vector is decomposed into two components, the absolute location of the closest point x on the torso surface, and the relative displacement vector to the torso surface v.

$$p = x + v \tag{1}$$

The avatar is calibrated to the subjects' height to minimize other distortions which would conflict with the visuo-proprioceptive conflict we are interested in.

**Movement distortion.** The subjects' arm movement is distorted by adding a constant offset between the real and the virtual hand to check if subjects are able to notice the discrepancy when fully extended or flexed. The distortion can be done through a modification of the relative surface displacement vector v in Eq 1 by using a distortion function D.

$$p = x + D(v) \tag{2}$$

The distortion model is a body-grounded function. It is defined using a constant offset $b$, added to the displacement vector $v$ along the normalized displacement vector $\hat{v}$ (Eq 3, see Fig 3). In this way, the distortion is the same for all experimental conditions (same reference, same amplitude, same direction, see Section Experimental procedure) and only the target position to reach varies across conditions.

$$D(v) = v + b\hat{v} \tag{3}$$

The offset component $b$ has been empirically chosen (5.5$cm$) so as to produce an unnoticeable constant discrepancy between the virtual hand and the real hand when reaching a target relatively close to the torso (arm flexed), but producing clearly distinct postures when reaching a target in full arm extension (e.g. the real arm is fully extended, but not the virtual arm). This assumption is checked against the answers of the questions asked during the experiment (see Section Experimental procedure).

## Experimental procedure

The experiment is divided into two blocks in a counterbalanced order (see Fig 2). The task is the same in the two blocks, only the questions are different. The first block is to assess the impact of the articular limits on the detection threshold. The second block is to assess the impact of the articular limits on body ownership, and thus on embodiment. Questions were asked after each trial. This procedure can thus reflect the experience of the subjects after each condition.

A constant position offset between the avatar's hand and participant's hand was used for the whole experiment. Only the target position and the direction of the distortion were changed (positive distortion = virtual hand ahead/negative distortion = virtual hand backward).

The experiment followed a 3x2 within-subject factorial design. The two main factors were the distortion magnitude (no distortion, negative distortion, positive distortion) and the arm posture (arm fully extended, flexed arm). Each condition was repeated six times in each block for a total of 36 trials per block. All subjects completed the two blocks. An explanation phase ensured that subjects understood the task, were not surprised by the distortion later on, and

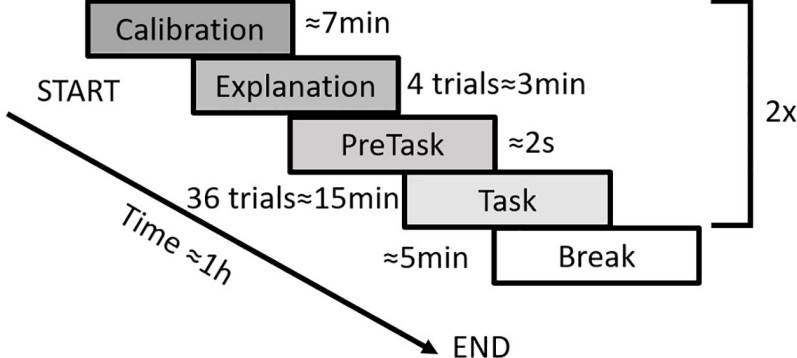

**Fig 2. Protocol overview.** After calibration, an explanation block provides instructions to the participants and is followed by a pre-task for training. The experimental task itself is repeated 36 times, consisting in a reaching movement followed by a questionnaire. After a short break, subjects repeat all steps of the block with the same task but a different questionnaire. The questionnaire consists in a research question and a control question (randomized). The research question is either on the detection of the posture mismatch (block 1) or on the self-attribution of the movement (block 2). This question is followed by a confidence rating.

knew how to answer questions; in practice subjects underwent two trials with no distortion and two trials with a positive distortion and a negative distortion. The distortion magnitude in those trials differed from the ones used during the experiment. Each block started with a short task (pretask on Fig 2) where subjects were asked to choose between two movements. The subjects had to fully extend their arms in both cases in order to reach the target. The two movements were the same as case 4 and case 6 (see Fig 3), except that the magnitude of the distortion was smaller. The experimental task itself was then repeated six times per condition, followed by a break of approximately five minutes. Subjects had to redo the calibration task after the break (possibly disrupted as they can stand up from the chair during the break) and redo the explanation task (questions are different in each block).

Subjects held a black cylinder in their right hand and a controller in their left hand. For every task, they started in the initial position i.e., the right hand near the torso and the left hand along the body. When the task started, a green cylinder appeared in front of them (Fig 4). They had to reach the green cylinder with their right hand so that the black cylinder fits inside. They had to wait until the green cylinder disappeared. As subjects looked at the cylinder while doing the task, the large horizontal FOV of the display ensured that the elbow of the right arm were always in the subjects' FOV. Then the scenes turned to black and the question was displayed in front of them in the virtual environment. To answer the question, they had to move a cursor horizontally with their head and validate the answer by pressing the button of the controller. Finally, the task was repeated until a message indicating the break appeared. During the break, they could remove the headset and stand up.

The distortion magnitude and the position of the cylinder varied at each trial, as shown in Fig 3. In cases 1, 3 and 5, the cylinder was in a position that did not require arm extension. Cases 2 and 4 required a physical full arm extension for the user. Case 6 required avatar's full arm extension (but user's flexion). When the distortion was present, its magnitude was the same, only the direction changed. When the distortion was negative (cases 3 and 4), subjects had to perform a movement with a bigger amplitude than normally required to reach the target. When the distortion was positive (cases 5 and 6), subjects had to shorten the movement amplitude. It is important to note that, in all the conditions, the virtual hand was always associated with the physical movement of the subject. Indeed, since the distortion introduced a discrepancy between the real hand of the user and the virtual hand of the avatar, the virtual hand was brought backward when the distortion amplitude was negative or was just brought forward when positive. For instance, during the full extension with a negative distortion, subjects were reaching the extension just before the avatar.

**Block 1: Detection threshold.** The goal of this block was to measure if subjects were able to detect the distortion in each trial. To this end, we asked the following forced choice Yes/No question, as done in similar previous studies [18, 22]:

*The arm posture corresponds exactly to mine.*

The detection rate is defined as the percentage of "No" for each condition. Block 1 thus aims at assessing whether hitting the articular limit enabled a better detection of the distortion, thus reducing their detection threshold. After this question, a second question was displayed to measure the subject's confidence in their answer. This question was rated on a continuous scale from 0:very unsure to 1:very sure (from [23]):

*Report your confidence on your previous response.*

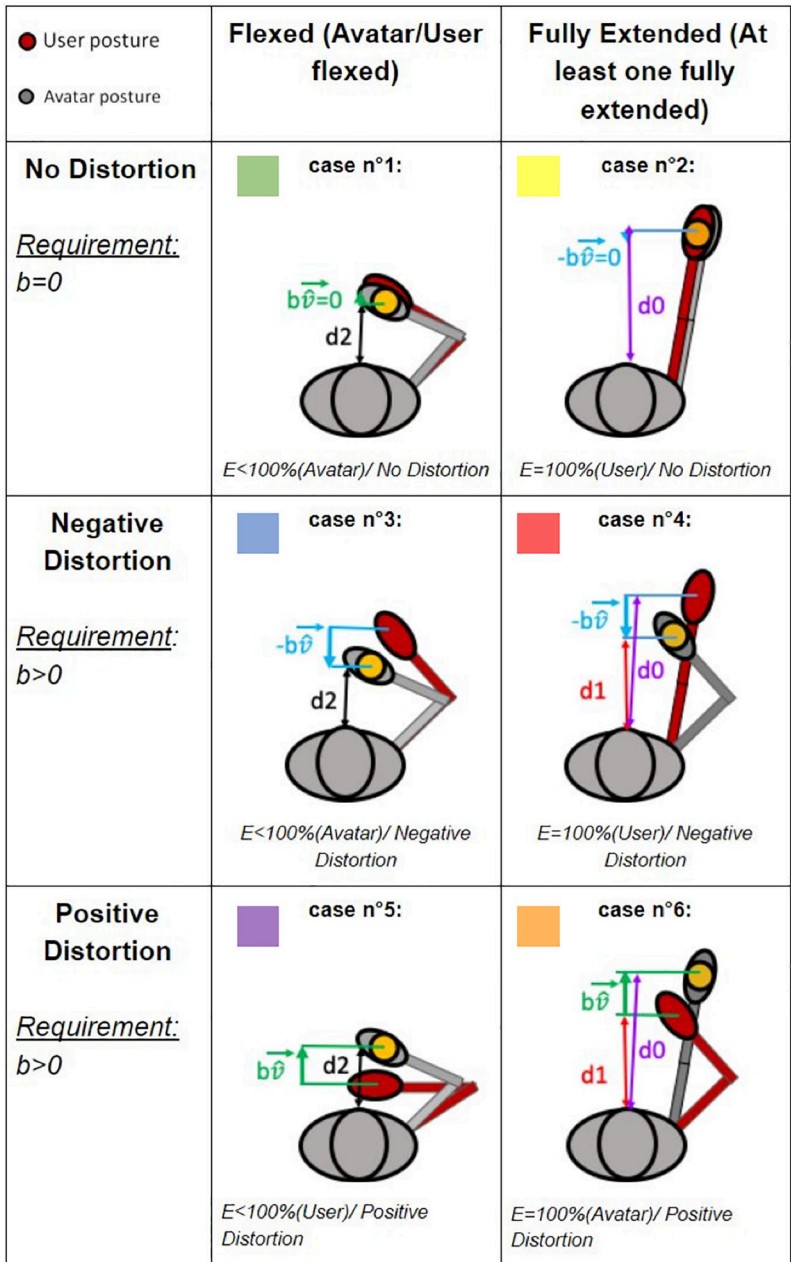

**Fig 3. Movement distortion in the six conditions.** The avatar is in grey, the subject in red, and the target in yellow; d0 represents the distance between the torso and the target when no distortion in the fully extended condition; d1 represents the distance between the torso and the hand for the flexed arm (virtual or real) when distortion in the fully extended condition; d2 represents the distance between the torso and the target when no distortion in the flexed condition. In flexed conditions, the avatar always covers the distance d2, contrary to the subject who covers a difference distance in distorted conditions. In full extension, either the avatar covers a shorter distance d1 than the subject who covers d0 (case 4), or the opposite happens (case 6).

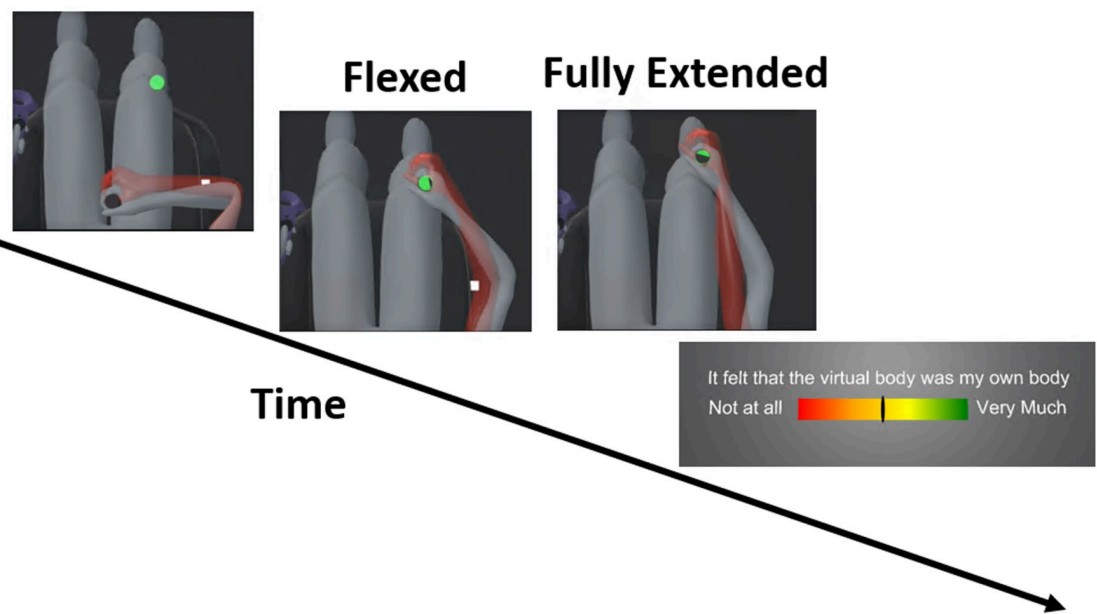

**Fig 4. The task.** Subjects held a black cylinder in their right hand and a controller in their left hand. Subjects started with their right hand near the torso and the left hand along the body. Once they had put the black cylinder inside the green cylinder, the green cylinder disappeared after few seconds. Then everything disappeared in the virtual room and the question was displayed. A cursor was controlled with their head and the answer was validated by pressing the button of the controller.

The confidence rating aims at revealing if some cases are more difficult to evaluate for subjects.

**Block 2: Self-attribution threshold.** The goal of this block was to measure subjects' body ownership for the virtual body during a trial. It is estimated by asking participants to rate the following statement on a continuous scale from 0:not at all to 1:very much (from [24]);

*It felt that the virtual body was my own body*

Although this question directly addresses only body ownership, it aims at deepening our knowledge about the impact of the articular limit on embodiment. We reason that, among the three components of embodiment [2], one is stable across all conditions through first-person perspective (self-location, [7, 25]), one is the independent variable manipulated with distortion (agency, [6]) and the third factor, body ownership, is of interest as it could be influenced by the detection of articular limit. Therefore, measuring body ownership would inform on the occurrence of breaks in embodiment when comparing conditions with identical agency (same distortion).

To allow us to validate our hypothesis on the embodiment question, we asked a control question. Participants had to rate the following statement on a continuous scale from 0:not at all to 1:very much (from [26]);

*It felt has if I had more than one body*

The two block-specific questions were always presented at the end of a task, in a randomized order.

## Statistical analysis

Statistical analysis was carried out using a non-parametric permutation test with a two-way repeated-measures analysis of variance (ANOVA), with our conditions as a fixed-effects factor and subjects as random effects. Differences were deemed statistically significant for p-values below the threshold $\alpha = .01$. We tested the assumption of the normality of residuals with the Kolmogorov-Smirnov test. Since they were not normal, we conducted a posthoc analysis using a non-parametric permutation test with a pairwise t-test for simple comparison, and Tukey's HSD test was used to correct for multiple comparisons. The effect size was computed using the scaled robust Cohen's standardized mean difference ($dr$) [27, 28] for non-normal residuals. The analysis was conducted using Matlab software. The demographic survey revealed only two persons with extensive experience in VR, four with good experience with VR, six with no experience and the remaining ones tried only few times. At the end of the experiment, we asked participants if they were able to see their elbow when immersed in VR; all confirmed that they could see their elbow during the task.

# Results

## Distortion threshold

The detection rate was higher when negative/flexed($M = 0.47$, $SD = 0.35$) compared to no distortion/flexed. The detection rate was also higher when negative/flexed compared to positive/flexed. These results show that the distortion is more easily perceived by users when the distortion is negative than when it is positive.

Moreover, the detection rate was higher when negative/fully extended ($M = 0.74$, $SD = 0.36$) compared to negative/flexed. Thus, the detection rate increased when subjects reached their articular limit while the avatar was flexed because of the negative distortion. This visuo-proprioceptive conflict had an impact on the detection threshold.

When looking at the significance levels, a main effect of conditions ($F(1, 24) = 566.2$, $p < 0.0001$) and an interaction effect ($F(5, 120) = 19.49$, $p < 0.0001$) were observed. The following effects resisted Post hoc analyses. The detection rate was significantly different ($t(24) = 5.15$, $p < 0.0001$) when negative/flexed compared to no distortion/flexed. The detection rate was also significantly different ($t(24) = 5.15$, $p < 0.0001$) when negative/flexed compared to positive/flexed.

Finally, the detection rate was significantly higher ($t(24) = 3.8$, $p < 0.01$) when negative/fully extended ($M = 0.74$, $SD = 0.36$) compared to negative/flexed. Thus, the detection rate increased significantly when subjects reached their articular limit while the avatar was flexed because of the negative distortion. This visuo-proprioceptive conflict had an impact on the detection threshold. However there was no significant difference($t(24) = 2.74$, $p < 0.076$) between positive/flexed and positive/fully extended ($M = 0.39$, $SD = 0.36$). When the avatar reached the articular limit while subjects were flexed because of the positive distortion, the detection rate increased but not significantly. For the same distortion magnitude, subjects noticed the distortion only if they reached their articular limit with their real body. If the avatar reached the articular limit but not the subjects with their real arms, they didn't notice the distortion.

## Confidence

Subjects seemed always confident in their answer, even in cases when it could have been expected that the question was difficult to answer (e.g. movement was distorted and a visuo-proprioceptive conflict occured).

When looking at the significance level, a main effect of conditions ($F(1, 24) = 1458.9$, $p < 0.0001$) was observed. However, post hoc analyses revealed no interaction effects.

This results confirm that the effects by this study were felt by subjects, and no case produced difficulties for subjects when answering.

## Body ownership

The score was lower when negative/flexed ($M = 0.66$, $SD = 0.22$) compared to no distortion/flexed. The negative distortion thus had an impact on the sense of embodiment but the body ownership score remained high.

The body ownership score was also higher when negative/flexed compared to when negative/fully extended ($M = 0.45$, $SD = 0.3$)(see Fig 5). The score for negative/fully extended was low. Thus, when subjects reached their articular limit and not the avatar's, the distortion tolerance was impacted. It reduced even further the embodiment score producing a BiE. The additional internal haptic feedback, when subjects reached their articular limit, was the source of the low body ownership score and seemed to dominate the visual representation of the arm posture.

When looking at the significance level, a main effect of conditions ($F(1, 24) = 722.9$, $p < 0.0001$) and an interaction effect ($F(5, 120) = 10.6$, $p < 0.0001$) were observed. The body ownership score was significantly higher when subjects were flexed.

The score was significantly different($t(24) = 3.68$, $p < 0.01$, $dr = 0.60$) when negative/flexed compared to no distortion/flexed. On the contrary the body ownership score was not significantly($t(24) = 1.61$, $p = 0.97$, $dr = 0.42$) different between positive/flexed ($M = 0.77$, $SD = 0.19$) and no distortion/flexed. Contrary to negative distortion the positive distortion did not not have any impact on the embodiment.

The body ownership score was significantly different ($t(24) = 3.8$, $p < 0.01$, $dr = 0.76$) when negative/flexed compared to when negative/fully extended. The body ownership score was not significantly higher ($t(24) = 3.0$, $p = 0.035$, $dr = 0.88$) when positive/flexed compared to when positive/fully extended ($M = 0.58$, $SD = 0.26$)(see Fig 5). Therefore, subjects tolerated the visuo-proprioceptive conflict when the subject's arm was not in full extension whereas the avatar's arm was. The lack of additional internal haptic feedback, when the avatar reached the

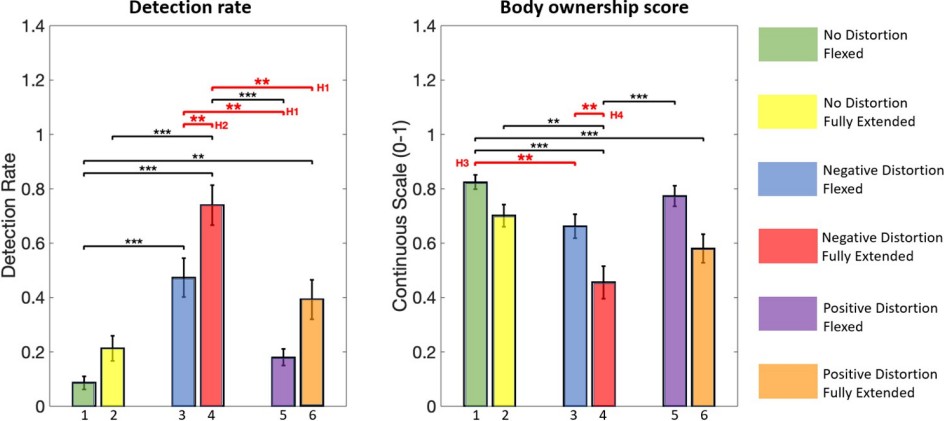

**Fig 5. Results for each experimental case (as numbered in Fig 3 with their respective color).** The detection rate was defined as the percentage of "No" for each condition to evaluate the impact of the articular limits on the detection threshold. Error bars represent the standard error of the mean. $^{**}p < 0.01$, $^{***}p < 0.001$.

articular limit, enabled the visual perception to dominate the proprioceptive cues provided by subjects' actual arm posture in the flexed posture.

## Control question

The scores for the control question were higher when there was distortion. The negative/fully extended ($M = 0.42$, $SD = 0.3$) was higher compared to no distortion/flexed ($M = 0.34$, $SD = 0.16$) and higher compared to positive/flexed ($M = 0.20$, $SD = 0.13$).

These results were unexpected. However, during the small debriefing at the end of the experiment, subjects reported that when the discrepancy was too high between their real arm and the virtual one, they felt like they were controlling two bodies, their real body and the virtual body. The negative/fully extended was the only case to produce such feeling completely disrupting the embodiment. Therefore, even if our control question is not a good control, the results do not contradict the effect on the body ownership scores.

These results are confirmed when looking at the significance level, a main effect of conditions ($F(1, 24) = 101.7$, $p < 0.0001$) and an interaction ($F(5, 120) = 5.98$, $p < 0.0001$) were observed. The negative/fully extended was significantly ($t(24) = -3.57$, $p < 0.01$, $dr = 0.82$) different from no distortion/flexed and significantly ($t(24) = 3.49$, $p < 0.01$, $dr = 0.71$)different from positive/flexed ($M = 0.20$, $SD = 0.13$).

## Discussion

Results first confirm a significantly higher detection rate when the distortion hinders the movement of participants. No significant difference between no distortion and positive distortion for the detection rate has been found in absence of visuo-proprioceptive conflict (Fig 5). Subjects did not perceive the distortion when they were helped as long as they did not reach their articular limit. On the contrary, subjects perceived the distortion when hindered even when there was no conflict. Thus, our first hypothesis (H1) is validated; negative distortion has more impact than positive distortion on the detection threshold. Secondly, the detection rate was significantly higher when subjects had their arm fully extended and not the avatar. Conversely, subjects didn't notice the opposite discrepancy, i.e. when the subject's arm was not in full extension whereas the avatar's arm was. The presence or absence of internal haptic proprioception might explain that subjects noticed more the distortion when they reached their own articular limit than merely observing the avatar fully extending the arm. These results partially validate our hypothesis (H2); visuo-proprioceptive conflicts (the subject reaches the articular limit) increase the detection rate, providing more insight on the importance of the proprioceptive feedback. Thirdly, we confirmed a significantly lower body ownership score when hindered than with no distortion. On the contrary, subjects showed a similar score when helped as with no distortion but only when no visuo-proprioceptive conflicts occurred. Interestingly, when there was distortion and no visuo-proprioceptive conflicts, embodiment scores remained above 50% and no break in embodiment was reported (even if subjects noticed the distortion). Subjects tolerated both negative and positive distortion as long as they did not reach an articular limit. These results partially validate our hypothesis (H3); negative distortion has more impact than positive distortion on the sense of embodiment only when no visuo-proprioceptive conflicts occurs. Finally, in the presence of a negative distortion, the embodiment score was significantly lower if there was a visuo-proprioceptive conflict than when there was none. The visuo-proprioceptive conflict was linked to embodiment scores below 60% only in one direction (negative) and may lead to a BiE. The impact in the positive direction was however not significant and less important, although the body ownership score was also below 60% and possibly also linked to a BiE. Our last hypothesis (H4) is therefore only partially

validated; the visuo-proprioceptive conflict induced by the articular limit disrupts the sense of embodiment only when subjects reach their articular limit and not the avatar's.

## Articular limit and internal haptic feedback

Our manipulation showed that an internal haptic feedback was experienced when subjects reached their own physical articular limit. Conversely, subjects tolerated the view of the avatar's arm being fully extended. As in Bovet et al. [16] for visuo-tactile conflict when touching the body, the visual dominance was disrupted and the participants rejected the virtual body as their body when an additional cue was provided to the subject (hitting their articular limit). According to Haggard et al. [29], a non-unified representation of the proprioceptive space reflecting the motor function might be the cause of such results. Subjects relied more on proprioception for depth and more on visual feedback for horizontal direction [30–32].

The subjects' peripersonal space (PPS) might also explain the subjects' sensitivity when reaching towards the articular limit. Previous studies [33, 34] showed that subjects more accurately detected a stimulation when reaching the boundary of their PPS. The results from other studies [35, 36] focused on interaction in VR also revealed that subjects produced fewer errors when interacting with a target placed at the limit of the reachable space. However, since subjects did not have any avatars, the sense of embodiment was not assessed. Since our study shows that subjects' sensitivity to the articular limit can break embodiment, it would be interesting to reproduce these previous works and study the impact of BiE in these experimental manipulations.

## Avatar representation, animation and calibration

As the representation of a virtual body of the user was progressively proven beneficial to the VR experience [37], a growing number of VR developers have integrated a 3D avatar animated with full-body motion capture [38], such as in location-based VR entertainment (e.g. The VOID, Dreamscape Immersive). The challenge however is that a poor calibration of the avatar can lead to visuo-proprioceptive conflicts, provoking BiE, and potentially negatively impacting the overall user experience. The current tendency of VR hardware heading towards large field of view displays (200˚ in this study) and ubiquitous motion capture systems also speak in favor of providing users with an avatar. But, as exemplified in this study, the user-avatar movement mapping should be done carefully, working with morphological as well as perceptual limitations.

Several methods were focused on calibrating the limb lengths and the shape of the avatar [39]. However, these studies did not focus on how the avatar's movements were perceived and if any visuo-proprioceptive conflict might occur and produce a BiE. We showed that subjects seemed particularly sensitive to how their (virtual) arm was animated when fully extended. Indeed, even though the difference was not statistically significant, we had an increase in the detection rate compared to the case where subjects flexed their arm with no distortion. This difference in the previous results might be due to anatomical differences. The motion range of the elbow can indeed differ according to gender, body mass index, sports practice, and subjects' inter-variability [40]. Thus, even with a precise calibration, the inverse kinematics algorithms do not currently integrate such a fine modeling of the anatomy for the animation of the fully extended arm. To conclude, the avatar has to be calibrated and animated so that the user never reaches full arm extension by surprise, with the avatar's arm still flexed, and favor the opposite if a perfect calibration is not possible.

## Embodiment

In our experiment, the sense of embodiment was induced by synchronising the subject's and the avatar's movements in a first person view. This active induction leads to a full-body ownership [6], which can override semantic incongruencies [41], or even lead to the experience of ownership for an invisible body [12]. Stern et al. [42] has shown that the sense of agency in this case is tied to the sense of body ownership. They distinguish it from other agency types by calling it *embodied sense of agency*. Importantly, Slater et al. [7] showed that, for the active induction to lead to body ownership, subjects had experienced a strong sense of agency for the avatar's movements. These results were further supported by the correlation found between the senses of agency and body ownership during active induction [43, 44]. We can therefore conclude from the high body ownership scores we observed that our participants experienced a strong sense of embodiment for their avatar, and that the significant drops in body ownership we observed for the same level of distortion also represented a decrease of their level of embodiment.

A distortion can indeed be tolerated in terms of subjective embodiment even when it is noticeable. As suggested by Gonzalez-Franco et al. [45], subjects embodied with an avatar adjusted to a continuous visuo-proprioceptive discrepancy. Kilteni et al. [46] even showed it is possible to provide a distorted feedback of the virtual arm, such as a very long upper limb, while providing synchronous multisensory correlations (e.g visuo-tactile feedback), and participants reported a high levels of ownership toward such distortion. Other studies [47, 48] showed that it is possible to provide body ownership illusions toward virtual bodies with distorted positions while providing synchronous visuo-tactile or visuo-motor correlations.

On the contrary, subjects are more likely to reject a discrepancy between their own body and their avatar if the disrupting event is apparent (hitting the arm articular limit in our case) by contradicting the motor plan (visual avatar not yet in full extension). Since the experience of embodiment is broken, the subjective experience of presence in the virtual environment might also be negatively impacted.

Finally, despite the visuo-proprioceptive conflict during movement distortions, subjects might tolerate the distortion more when a positive distortion helped them to execute the movement. On the contrary, a negative distortion preventing subjects from achieving their task have more impact on the detection threshold [49]. We confirm and extend these results; only in the condition when the distortion was negative, the internal feedback provided by the articular limit seemed to be the breaking point which makes subjects aware of the distortion.

## Conclusion

We have expanded the general knowledge about the impact of an articular limit on the detection threshold and the sense of embodiment and could provide important knowledge about the interaction between visual perception and proprioception. We have also completed previous works by studying the distortion direction and the reach of an articular limit within the same experiment, revealing the interaction effect on the SoE.

We found that the resulting types of animation errors were perceived differently. Indeed, if users felt the internal haptic feedback while the avatar's arm was still flexed, they perceived the visuo-proprioceptive conflict and rejected the avatar as their body. On the contrary, if users did not perceive any internal haptic feedback, even if the avatar reached its own articular limit, they kept feeling embodied inside the avatar. This new knowledge could be used to better design animation and help to choose the best algorithms when animation errors cannot be avoided. For instance, the calibration of the limb length and the position of the join center of an avatar is not an easily resolved problem. Therefore, some errors of calibration may remain

during the animation. However, one might push toward one type of error, the one tolerated by users, and choose to have the virtual limbs slightly shorter than the real ones. This will prevent the visuo-proprioceptive conflict due to the user's full arm extension. Hopefully, these results will help future developers and the research community when designing a new interaction for full-body animation.

## Acknowledgments

We would like to thank Ms. Francesca Gieruć and M. Mathias Delahaye for their invaluable contributions.

## Author Contributions

**Conceptualization:** Thibault Porssut, Olaf Blanke, Ronan Boulic.

**Data curation:** Thibault Porssut, Bruno Herbelin, Ronan Boulic.

**Formal analysis:** Thibault Porssut.

**Funding acquisition:** Bruno Herbelin, Ronan Boulic.

**Investigation:** Thibault Porssut, Olaf Blanke, Bruno Herbelin, Ronan Boulic.

**Methodology:** Thibault Porssut, Olaf Blanke, Bruno Herbelin, Ronan Boulic.

**Project administration:** Olaf Blanke, Ronan Boulic.

**Resources:** Thibault Porssut, Bruno Herbelin, Ronan Boulic.

**Software:** Thibault Porssut.

**Supervision:** Bruno Herbelin, Ronan Boulic.

**Validation:** Thibault Porssut, Bruno Herbelin, Ronan Boulic.

**Visualization:** Thibault Porssut, Bruno Herbelin, Ronan Boulic.

**Writing – original draft:** Thibault Porssut, Bruno Herbelin, Ronan Boulic.

**Writing – review & editing:** Thibault Porssut, Olaf Blanke, Bruno Herbelin, Ronan Boulic.

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
