## [Decision Letter · Decision Letter 0]

16 Dec 2021

PONE-D-21-23301Reaching articular limits can negatively impact embodiment in virtual reality.PLOS ONE

Dear Dr. Porssut,

Thank you for submitting your manuscript to PLOS ONE. After careful consideration, we feel that it has merit but does not fully meet PLOS ONE’s publication criteria as it currently stands. Therefore, we invite you to submit a revised version of the manuscript that addresses the points raised during the review process.

We look forward to receiving your revised manuscript.

Kind regards,

Imre Cikajlo, Ph.D.

Academic Editor

PLOS ONE

“We would like to thank Ms. Francesca Gieru´c and M. Mathias Delahaye for their invaluable contributions. This work has been supported by the SNFS project ’Immersive Embodied Interactions’ grant 200020 178790.

“T.P

This work has been supported by the SNFS project 'Immersive Embodied Interactions' grant 200020_178790.

Swiss National Science Foundation:

https://www.snf.ch/en

NO: The funders had no role in study design, data collection and analysis, decision to publish, or preparation of the manuscript.”

Additional Editor Comments:

Please read carefully the remarks of the reviewers and take care of the minor issues.

Reviewers' comments:

Reviewer's Responses to Questions

**Comments to the Author**

1. Is the manuscript technically sound, and do the data support the conclusions?

Reviewer #1: Yes

Reviewer #2: Yes

2. Has the statistical analysis been performed appropriately and rigorously? 

Reviewer #1: Yes

Reviewer #2: Yes

3. Have the authors made all data underlying the findings in their manuscript fully available?

Reviewer #1: No

Reviewer #2: Yes

4. Is the manuscript presented in an intelligible fashion and written in standard English?

Reviewer #1: No

Reviewer #2: Yes

5. Review Comments to the Author

Reviewer #1: This paper reports an experiment where different configurations of the real and virtual arm in outstretched position are examined for their effects on body ownership of the virtual body. The experiment is well described and the statistical analysis is excellent. There are a few points that could improve the paper.

Although the submission form states that the data is fully available, I could not see where this was indicated in the paper - i.e., from where the data can be accessed.

In 'Equipment and software' please give the weight of the head mounted display.

In the next paragraph it is not clear whether it is only the virtual hand that holds the cylinder or whether subjects were also holding a cylinder. This seems to be cleared up later in the figure caption - that they were actually holding the cylinder. Was the cylinder tracked, or were transformations to the cylinder computed from transformations to the hand orientation and translation?

Some of the English reads a bit strangely and the paper should be thoroughly checked for this. For example, "All subjects have been through the two blocks" rather than "All subjects completed the two blocks".

"Although this question directly addresses only body ownership, it aims at informing on the impact of the articular limit on embodiment."

" ... at informing on ..." doesn't seem to be a correct formulation.

"subjects shall experience a strong sense of agency for the avatar's movements" in the discussion - the "shall" makes it seem like an imperative.

Figure 5 - the questionnaire scores run from 0 to 1 but the vertical axes extend to 1.4. This is confusing and must be corrected. I think that all the significance *s above the bars need to be pruned, there are too many and the graph becomes difficult to read. Please only include the critical ones that relate directly to the hypotheses.

From the graph it is clear without any significance tests that the critical impact is associated with "Negative Distortion Fully Extended" and "Positive Distortion Fully Extended" which both result in an obvious reduction in body ownership compared to the others. All of the others, even though there may be some significant differences between them, show relatively high levels of body ownership. So "fully extended" with any kind of distortion leading to lower body ownership is the main finding. Although there are differences between the other conditions can the authors please mention also the effect sizes of these differences rather than only significance levels.

In the Results first paragraph I think it would be helpful to readers to first discuss the findings from the graphs in terms of what seems to be greater than what before rushing into discussion of significance levels without explaining what these mean. E.g., we can be told that "x is significantly different from y" without knowing whether this indicates whether x > y or y > x. So it may be preferable to give the latter information first, and then discuss the significance.

In the discussion it says that "subjects severely reject a discrepancy between their own body". But looking at the raw data 'severely reject' seems to be an over statement. It seems that only in the case of "fully extended" with any kind of distortion is there a clear drop in body ownership compared to the other conditions. All of the others are way above the 0.5 mark on the questionnaire scale.

In the discussion "subjects might be more willing not to notice the distortion" - "willing" makes it seem like this is a conscious deliberate choice on the part of the subjects.

Overall this is an excellent and useful contribution to the area of virtual embodiment, and will be important for researchers in this field.

Reviewer #2: The manuscript is well written and the study design is well done. The results from this study can be helpful for conduction virtual reality expriments with healthy and clinical populations including embodied conditions. However the authors should consider some aspects for improving the quality of the manuscript.

- Introduction section:

line 3: add a reference aafter SoE.

Suggestion for improving definitions of self-location, agency and body ownership:

Sense of Self-location: Self-location is a determinate volume in space where one feels to be located. Normally self-location and body-space coincide in the sense that one feels self-located inside a physical body (Lenggenhager, Mouthon, & Blanke, 2009).

Sense of Agency: The sense of agency refers to the sense of having ‘‘global motor control, including the subjective experience of action, control, intention, motor selection and the conscious experience of will’’ (Blanke & Metzinger, 2009, p. 7).

Sense of Ownership: Body ownership refers to one’s self-attribution of a body (Gallagher, 2000; Tsakiris, Prabhu, & Haggard, 2006).

Third pagraph: the authors say 'we identified a relatively frequent problem likely to interrupt the

visuo-proprioceptive integration' ,where you identified this problem? Is there any study arguing this situation? If yes, please cite it.

End of the 4th paragraph: I agree with the impact of internal haptic feedback on movement perception in VR, however the authors should consider that if reserachers provide strong sense of agency making participants actively participate in the initiation of the virtual body movement or vistual body action, this may descrease or delete the mistmatch conflicts, and providing synchronous visuo-tactile correlations. See Kokkinara, E., Kilteni, K., Blom, K. J., & Slater, M. (2016). First person perspective of seated participants over a walking virtual body leads to illusory agency over the walking. Scientific reports, 6

Participants section: not controlling the dominance laterality should be added as a limitation of the study. Even though the difuclty of the task, the joint receptors of the participants will be more sensitive, it should be a higher proprioceptive sense in the dominant upper limb compared to the non-dominant upper limb.

Discussion: the authors should consider that there are evidence showing that it is poosible to provide a distorted fedback of the virtual arm, such as an hyperextension of the upper limb, while providing synchronous multisensory correlations (e.g visuo-tactile feedback), and participants reported higher levels of wnership toward such distortion:

- Kilteni, K., Normand, J. M., Sanchez-Vives, M. V., & Slater, M. (2012). Extending body space in immersive virtual reality: a very long arm illusion. PloS one, 7(7), e40867.

Further, other studies showed that it is possible to provide body ownership illusions toward virtual bodies with distorted positions while providing synchronous visuo-tactile or visuo-motor correlations:

-Bergström, I., Kilteni, K., & Slater, M. (2016). First-person perspective virtual body posture influences stress: a virtual reality body ownership study. PloS one, 11(2), e0148060.

- Matamala-Gomez, M., Nierula, B., Donegan, T., Slater, M., & Sanchez-Vives, M. V. (2020). Manipulating the perceived shape and color of a virtual limb can modulate pain responses. Journal of clinical medicine, 9(2), 291

Avatar representation section: Change the format about the take a home message. I suggest the authros to re-formulate a sentence as an statment.

Conclusion section can be shortened.

6. PLOS authors have the option to publish the peer review history of their article (what does this mean?). If published, this will include your full peer review and any attached files.

Reviewer #1: No

Reviewer #2: **Yes: **Marta Matamala-Gomez

---

## [Author Response · Author response to Decision Letter 0]

8 Jan 2022

We would like to thank the reviewers for their helpful feedbacks. We have included with our revised submission a document entitled "Response to Reviewers" with all the details of all the modifications made in the paper and detailed answers of each specific reviewer comment.

Thank you in advance for your help,

Best Regards,

Thibault Porssut

---

## [Decision Letter · Decision Letter 1]

2 Feb 2022

Reaching articular limits can negatively impact embodiment in virtual reality.

PONE-D-21-23301R1

Dear Dr. Porssut,

We’re pleased to inform you that your manuscript has been judged scientifically suitable for publication and will be formally accepted for publication once it meets all outstanding technical requirements.

Kind regards,

Imre Cikajlo, Ph.D.

Academic Editor

PLOS ONE

Additional Editor Comments (optional):

Reviewers' comments:

Reviewer's Responses to Questions

**Comments to the Author**

1. If the authors have adequately addressed your comments raised in a previous round of review and you feel that this manuscript is now acceptable for publication, you may indicate that here to bypass the “Comments to the Author” section, enter your conflict of interest statement in the “Confidential to Editor” section, and submit your "Accept" recommendation.

Reviewer #1: All comments have been addressed

Reviewer #2: All comments have been addressed

2. Is the manuscript technically sound, and do the data support the conclusions?

Reviewer #1: (No Response)

Reviewer #2: Yes

3. Has the statistical analysis been performed appropriately and rigorously? 

Reviewer #1: (No Response)

Reviewer #2: Yes

4. Have the authors made all data underlying the findings in their manuscript fully available?

Reviewer #1: (No Response)

Reviewer #2: Yes

5. Is the manuscript presented in an intelligible fashion and written in standard English?

Reviewer #1: (No Response)

Reviewer #2: Yes

6. Review Comments to the Author

Reviewer #1: (No Response)

Reviewer #2: (No Response)

7. PLOS authors have the option to publish the peer review history of their article (what does this mean?). If published, this will include your full peer review and any attached files.

Reviewer #1: No

Reviewer #2: **Yes: **Marta Matamala-Gomez

---

## [Editor Report · Acceptance letter]

11 Feb 2022

PONE-D-21-23301R1 

Reaching articular limits can negatively impact embodiment
in virtual reality. 

Dear Dr. Porssut:

I'm pleased to inform you that your manuscript has been deemed suitable for publication in PLOS ONE. Congratulations! Your manuscript is now with our production department. 

Kind regards, 

on behalf of

Professor Imre Cikajlo 

Academic Editor

PLOS ONE